# Turning Points in Cross-Disciplinary Perspective of Primary Hyperparathyroidism and Pancreas Involvements: Hypercalcemia-Induced Pancreatitis, *MEN1* Gene-Related Tumors, and Insulin Resistance

**DOI:** 10.3390/ijms25126349

**Published:** 2024-06-08

**Authors:** Mara Carsote, Claudiu Nistor, Ana-Maria Gheorghe, Oana-Claudia Sima, Alexandra-Ioana Trandafir, Tiberiu Vasile Ioan Nistor, Bianca-Andreea Sandulescu, Mihai-Lucian Ciobica

**Affiliations:** 1Department of Endocrinology, “Carol Davila” University of Medicine and Pharmacy, 020021 Bucharest, Romania; carsote_m@hotmail.com; 2Department of Clinical Endocrinology V, “C.I. Parhon” National Institute of Endocrinology, 011863 Bucharest, Romania; 3Department 4-Cardio-Thoracic Pathology, Thoracic Surgery II Discipline, “Carol Davila” University of Medicine and Pharmacy, 050474 Bucharest, Romania; 4Thoracic Surgery Department, “Dr. Carol Davila” Central Military University Emergency Hospital, 010242 Bucharest, Romania; 5PhD Doctoral School, “Carol Davila” University of Medicine and Pharmacy, 020021 Bucharest, Romania; ana-maria.gheorghe@drd.umfcd.ro (A.-M.G.); oana-claudia.sima@drd.umfcd.ro (O.-C.S.); alexandra-ioana.trandafir@drd.umfcd.ro (A.-I.T.); bianca-andreea.sandulescu@drd.umfcd.ro (B.-A.S.); 6Department of Clinical Biochemistry, “Iuliu Hatieganu” University of Medicine and Pharmacy, 400347 Cluj-Napoca, Romania; 7Department of Internal Medicine and Gastroenterology, “Carol Davila” University of Medicine and Pharmacy, 020021 Bucharest, Romania; lucian.ciobica@umfcd.ro; 8Department of Internal Medicine I and Rheumatology, “Dr. Carol Davila” Central Military University Emergency Hospital, 010825 Bucharest, Romania

**Keywords:** gene, neuroendocrine, insulin, pancreas, calcium, parathyroid, surgery, parathormone, metabolic

## Abstract

We aimed to provide an in-depth analysis with respect to three turning points in pancreas involvement in primary hyperparathyroidism (PHP): hypercalcemia-induced pancreatitis (HCa-P), MEN1 (multiple endocrine neoplasia)-related neuroendocrine tumors (NETs), and insulin resistance (IR). This was a comprehensive review conducted via a PubMed search between January 2020 and January 2024. HCa-P (*n* = 9 studies, N = 1375) involved as a starting point parathyroid NETs (*n* = 7) or pancreatitis (*n* = 2, N = 167). Case report-focused analysis (N = 27) showed five cases of pregnancy PHP-HCa-P and three reports of parathyroid carcinoma (female/male ratio of 2/1, ages of 34 in women, men of 56). MEN1-NET studies (*n* = 7) included MEN1-related insulinomas (*n* = 2) or MEN1-associated PHP (*n* = 2) or analyses of genetic profile (*n* = 3), for a total of 877 MEN1 subjects. In MEN1 insulinomas (N = 77), the rate of associated PHP was 78%. Recurrence after parathyroidectomy (N = 585 with PHP) was higher after less-than-subtotal versus subtotal parathyroidectomy (68% versus 45%, *p* < 0.001); re-do surgery was 26% depending on surgery for pancreatic NETs (found in 82% of PHP patients). *MEN1* pathogenic variants in exon 10 represented an independent risk factor for PHP recurrence. A single pediatric study in MEN1 (N = 80) revealed the following: a PHP rate of 80% and pancreatic NET rate of 35% and 35 underlying germline *MEN1* pathogenic variants (and 3/35 of them were newly detected). The co-occurrence of genetic anomalies included the following: *CDC73* gene variant, glucokinase regulatory protein gene pathogenic variant (c.151C>T, p.Arg51*), and CAH-X syndrome. IR/metabolic feature-focused analysis identified (*n* = 10, N = 1010) a heterogeneous spectrum: approximately one-third of adults might have had prediabetes, almost half displayed some level of IR as reflected by HOMA-IR > 2.6, and serum calcium was positively correlated with HOMA-IR. Vitamin D deficiency was associated with a higher rate of metabolic syndrome (*n* = 1). Normocalcemic and mildly symptomatic hyperparathyroidism (*n* = 6, N = 193) was associated with a higher fasting glucose and some improvement after parathyroidectomy. This multilayer pancreas/parathyroid analysis highlighted a complex panel of connections from pathogenic factors, including biochemical, molecular, genetic, and metabolic factors, to a clinical multidisciplinary panel.

## 1. Introduction

Primary hyperparathyroidism, an endocrine condition causing hypercalcemia due to parathormone (PTH) producing single (accounting 80–85% of all cases) or multiple parathyroid tumors, stands for a myriad of acute and chronic clinical elements such as kidney stones (and eventually acute or chronic renal failure), bone loss and associated fragility fractures, and arrhythmia, as well as non-specific features such as dry skin, impaired sleep, headache, chest pain, fatigue/asthenia, polyuria/polydipsia, etc. [1,2].

Currently, a normocalcemic or mildly symptomatic presentation is more frequently registered than the traditional picture of longstanding manifestations or hypercalcemic crisis due to early detection in many countries [3,4,5]. Females are two to three times more prone for the condition than males, particularly middle-aged women, but children and teenagers might be affected too, the majority having a more severe phenotype when compared to adults [6,7].

Regarding the digestive panel (including pancreatic insulin profile), several clusters should be specified: firstly, anorexia, nausea, and vomiting may be a consequence of PTH-dependent hypercalcemia, a gastro-duodenal ulcer or pancreatitis, or a connected complaint related to nephrolithiasis [8]. Successful parathyroidectomy and consecutive calcium level correction involve a massive clinical improvement in the digestive panel, as similarly seen in most clinical aspects, noting that parathyroid surgery represents the single curative approach with a general rate of success of 90–97% depending on the study [9,10].

Secondarily, the co-presence of gastric, duodenal, and pancreatic neoplasia, in the majority of neuroendocrine tumors (NETs), represents a syndromic consequence via a common genetic background in multiple endocrine neoplasia (MEN) syndrome that comes with an increased burden of disease due to synchronous or asynchronous neoplasms requiring multidisciplinary management [11,12,13].

Thirdly, another important chapter is the higher risk of cardio-metabolic traits underlying insulin resistance in adults with primary hyperparathyroidism; hence, anomalies of pancreas-related insulin secretion and its peripheral receptors’ dysregulation represent one more milestone in the issue of pancreas findings (of functional type in this particular instance, not of tumor type as seen in pancreatic NETs such as insulinomas) in the parathyroid field [14,15].

Across this comprehensive review, we aimed to provide an in-depth analysis with respect to various pancreas involvements in patients diagnosed with parathyroid NETs, particularly those functioning types causing primary hyperparathyroidism. Specifically, there are three main aspects to present: hypercalcemia that causes pancreatitis amid these parathyroid tumors, pancreatic and parathyroid NETs’ co-presence, and the pancreas’s involvement as an essential contributor to insulin resistance that is identified in subjects with primary hyperparathyroidism. This is a narrative overview of the literature introducing a multilayer-structured analysis across a multidisciplinary approach. We searched full-length, freely accessible, English papers across the PubMed database (between January 2020 and January 2024) by using different combinations of keywords (within the title and/or abstract) such as “pancreas” (alternatively, “pancreatitis”, “insulin”, “insulin resistance”, “diabetes”, or “metabolic syndrome”) and “parathyroid” (alternatively, “calcium”, “hypercalcemia”, “primary hyperparathyroidism”, or “MEN”) (Figure 1).

### 1.1. Pancreatitis and PTH-Dependent Hypercalcemia

As mentioned, a large spectrum of digestive complaints might be described when it comes to primary hyperparathyroidism which otherwise stands for a classical endocrine pathology. Despite the fact that these are non-specific, a certain index of suspicion should be taken into consideration across daily gastroenterological assessment. For instance, gastroesophageal reflux may be related to the hypercalcemia-associated direct effects or to MEN1 components [16], but, also, acute lumbar pain with abdominal spreading may be caused by the presence of kidney and ureteral stones [17]. A review published in 2021 focusing on gastrointestinal symptoms in primary hyperparathyroidism found (N = 331 patients coming from *n* = 13 studies published between 2007 and 2020) that these complaints represented the fourth most common cause of presentation in the parathyroid condition (including abdominal pain in 18.67% of the subjects) [18]. Moreover, in the pediatric population, abdominal pain may be described in 11–45% of children and teenagers at one point underlying a wide panel of conditions, while primary hyperparathyroidism remains rare in this age group; the pediatric subgroup displays in 30–87% of patients’ various digestive complaints [18].

Acute or chronic pancreatitis massively impacts parathyroid disease prognosis due to high morbidity and mortality [19]. Typically, the levels of calcium are low in pancreatitis of other causes (with intact parathyroid glands); that is why the identification of a high serum calcium value (or even inadequately normal blood calcium) remains the most valuable clue of further performing the specific parathyroid-related hormonal panel. Recurrent abdominal pain should also be an additional clue for checking PTH. Nevertheless, PTH might be found to be elevated (as suggestive for the biological confirmation of primary hyperparathyroidism) or suppressed (as seen in hypercalcemia of malignancy which is associated with cancers of different origins). Neither of these two entities stands for a frequent cause of pancreatitis in the general population, hypercalcemia accounting for less than 1% of all cases (some reports showed a rate between 1.5% and 8%) [20]. Of note, in 2021, an exceptional case of pancreatitis synchronously presenting both mechanisms of hypercalcemia (PTH-dependent and -independent) was reported [21].

The exact underlying mechanisms of pancreatitis in primary hyperparathyroidism are still a matter of debate [22,23,24]. Some authors appreciate that high serum calcium contributes to increased local calcium in pancreas cells causing the micro-aggression of the ducts and parenchyma; others consider that hypercalcemia actually reduces pancreatic secretion but does not influence the enzymes’ capacity; thus, a protein’s coagulation is induced, and this has the capacity to seal the pancreatic ducts with consecutive obstruction [22].

Clinically, hypercalcemia-related pancreatitis is associated with a presentation that is not specific for this pathogenic type (upper abdominal pain/discomfort, nausea, vomiting, jaundice, bloating, asthenia, reduced appetite, weight loss, etc.) and provides little information for performing endocrine assays [22,23]. In many cases, the originating primary parathyroid NET might not be obvious at first presentation for pancreatitis, meaning that PTH-derivate hypercalcemia may act completely asymptomatic (or unrecognized) until the episode of pancreatic involvement [25]. As a practical point, identifying an elevated blood calcium level at the moment of pancreatitis diagnosis requires an additional PTH assessment.

In short, according to our methods, we identified nine studies to address the presence of pancreatitis in primary hyperparathyroidism, having as a starting point a population confirmed with parathyroid NETs (*n* = 7) or different types of pancreatitis (*n* = 2) [20,26,27,28,29,30,31,32]. Seven studies were retrospective [20,27,28,29,30,32], one was prospective [26], and another was a registry-based cohort [31], for a total of 1375 individuals. The sample size in cohorts primarily confirmed with acute or chronic pancreatitis was 67 [26] of 100 persons [20], while within the studies firstly addressing subjects with primary hyperparathyroidism, there were 35 patients (of note, this was the single study on the pediatric population) [29], 30 persons (compared with 30 hyperparathyroidism-free controls) [30], and 51 individuals [27], for, respectively, larger cohorts of 242 [32], 386 (particularly gestational hyperparathyroidism) [31], and 464 [28] (Table 1).

A study on acute pancreatitis in patients diagnosed with parathyroid NETs was published in 2022 by Rashmi et al. [27]; this was a retrospective analysis of 51 patients who were admitted at a single tertiary center between 2010 and 2021. Among these, 23.52% experienced any type of pancreatitis (12/51), namely 9.8% (N1 = 5/51) had an acute form, and 13.73% (N2 = 7/51) had chronic pancreatitis. Acute episode diagnosis was established based on having at least two elements from the following three: abdominal pain, increased blood amylase or lipase for a minimum of three times, or suggestive imaging findings for pancreatitis. Patients with primary hyperparathyroidism who experienced an episode of acute pancreatitis (5/51) were more frequently males, and their age of onset was decreased when compared to subjects who did not experience this type of pancreatic involvement during hypercalcemia (35.20 ± 16.11 versus 49.23 ± 14.80 years, *p* = 0.05). Apparently paradoxically, they had lower serum PTH (*p* = 0.01) but similar blood calcium levels (*p* = 0.32). As expected, this subgroup more often presented gastrointestinal complaints (pain, nausea, and vomiting) but not more often traditional renal and osseous complications of the endocrine condition [27]. This suggests that a particular set of patients might be prone to pancreatic complications, rather than having kidney stones or secondary osteoporosis and fragility fractures, or, on the other hand, that the presence of symptomatic pancreas anomalies might lead to the early confirmation of high PTH-related hypercalcemia. Further on, once identified, the early removal of parathyroid NETs might avoid long-term non-digestive complications of primary hyperparathyroidism [27,33].

On the other hand, another study showed that the rate of kidney stone confirmation is higher in patients with parathyroid NET-related pancreatitis, suggesting potential common pathogenic elements with concern to high calcium effects. This was a retrospective study on 30 patients admitted for acute pancreatitis caused by primary hyperparathyroidism as the first clinical element compared with 30 individuals diagnosed with the same parathyroid condition but without any pancreas involvement. Arora et al. [30] identified a higher rate of nephrolithiasis within the first group (having an average age at pancreatitis diagnosis of 44.9 ± 13.9 years and a male-to-female ratio of 1.3), a lower rate of bone complications (as prior mentioned data [27]), and higher serum calcium versus controls. Of note, one patient posed an additional severity due to overlapping pancreatitis and chronic kidney failure, one of the renal complications of longstanding uncontrolled hypercalcemia [30].

Besides the anomalies of the calcium profile, the presence of complicated diabetes mellitus at the moment of hypercalcemia-associated acute pancreatitis, including ketoacidosis (which was reported in two adults according to our research), might precipitate the early detection of parathyroid tumors [34,35]. Kumari et al. [28] showed among 464 subjects with primary hyperparathyroidism that the type 2 diabetic subgroup (N = 54) had a higher prevalence of pancreatitis (22.2%) when compared to the nondiabetic subgroup (5.6%, *p* = 0.07) [28]. This suggests a potential interplay between the PTH/calcium profile and insulin secretion/peripheral actions, as we will mention below.

Another single-center study conducted by Tiwari et al. [20] and published in 2022 showed in 100 people confirmed with acute pancreatitis of different causes (between January 2021 and December 2021) that the hypercalcemia-related pathogenic type was found in 3% (N = 3) of them. One out of these three individuals had primary hyperparathyroidism due to a parathyroid adenoma. The overall mortality rate was 11% [20].

A particular entity is represented by acute pancreatitis with gestational primary hyperparathyroidism (representing less than 1–2% of all female cases that present a parathyroid NET) [31]. Acute severe hypercalcemia complicated or not [36] with acute pancreatitis has been reported in gestation as the first diagnosis [37,38,39], particularly in young females associated with genetic forms of hyperparathyroidism [40]. Sometimes, acute hypercalcemia triggers or mimics eclampsia/pre-eclampsia and complicates maternal/fetal outcomes, while parathyroidectomy might be necessary during pregnancy if medical therapy fails to control the abnormal serum calcium levels; alternatively, in other cases, an emergency cesarean is performed, and postpartum parathyroidectomy is later provided [38]. We identified five single case reports of this specific circumstance (age at presentation of 26, 30, 31—for two patients—and 40 years; two of these women developed postpartum pancreatitis [38,41]) [37,38,39,41,42] and one of the previously mentioned studies [31]. For example, Dias Leite et al. [41] reported on a 40-year-old primigravida who was admitted as an emergency case for pre-eclampsia at week 27 of gestation. Due to the severity of this episode, pregnancy termination via a cesarean was mandatory, but it was followed by postpartum hypercalcemia-related acute pancreatitis caused by two synchronous parathyroid adenomas [41]. Moreover, a study based on the Indian registry of primary hyperparathyroidism between 2005 and 2020 showed that among 386 women diagnosed with the condition, 2.1% experienced a gestational form; half of this subgroup had acute pancreatitis as a common clinical finding. As a general note, the rate of gestational hyperparathyroidism in this study was double when compared to most studies on the general population [31,43,44] (Table 2).

An extremely high calcium concentration causing acute pancreatitis may be induced by a parathyroid carcinoma, a dramatic orphan endocrine malignancy [45,46]. In 2021, a novel adult case was reported showing a misdiagnosis of pancreatic malignancy due to distant metastases from a parathyroid carcinoma complicated with acute pancreatitis. Mignini et al. [47] provided the results of a PubMed search between 1969 and 2021 with respect to parathyroid cancer-related acute pancreatitis and identified only twelve prior cases [47]. Moreover, a new case was reported in 2022 by Zelano et al. [48] in a 56-year-old male with moderate pancreatitis and multiple non-digestive complications [48]. On an interesting note, in 2020, another challenging case of a lady who experienced recurrent acute pancreatitis amid recurrent primary hyperparathyroidism following initial surgery was revealed with the concern of a very rare location, namely a mediastinal parathyroid carcinoma that required additional interventions. Jiajue et al. [49] revised prior published data on this particular site of the parathyroid malignancy and identified twenty-one cases reported between 1959 and 2019, regardless of the complications’ panel [49].

We summarized a total of three single case reports of pancreatitis caused by a parathyroid malignancy (female-to-male ratio of one to two; age of 56 years for men, 34 years in women with mediastinal location) [47,48,49] (Table 3). 

Chronic pancreatitis, typically related to chronic alcohol intake or gallstones but also severe hypertriglyceridemia or unusual causes such as the, recently identified, cannabis-induced type [50], may be the effect of longstanding PTH-dependent hypercalcemia. The connection between high PTH concentration in addition to consecutive elevated serum calcium and pancreatitis was first established by Cope in 1957 who published two cases; however, the first such case dates from 1903 (this was a postmortem diagnosis) [22,51,52].

Calcific pancreatitis may be regarded as a sub-type of chronic pancreatic involvement. In 2020, the first case of MEN-associated calcific pancreatitis was reported in a 52-year-old woman with MEN1 (parathyroid NET was associated with a pituitary NET and a corticoadrenal adenoma) [53]. Also, we mention a single-centric observational prospective study from 2023 on patients with chronic painful calcific pancreatitis (N = 67) who underwent endoscopic retrograde cholangiopancreatography to release the pain and were followed up for two years; during this analysis, the authors identified a novel subject with primary hyperparathyroidism, representing 1.5% of the entire cohort [26].

Usually, an acute episode of pancreatitis is not recognized as being triggered by primary hyperparathyroidism, and patients continue to have two or three episodes until adequate identification of the underlying parathyroid condition [22,54]. For instance, Mehta et al. [55] reported the case of an adult lady who suffered from three episodes of acute pancreatitis during six months in the absence of alcohol consumption or gallstones until parathyroid-related hypercalcemia was incriminated as the pancreatic disease [55].

Moreover, a single-centric retrospective study conducted between 1989 and 2019 in 35 pediatric subjects (average age of 15.2 ± 2.9 years) diagnosed with primary hyperparathyroidism (8.5% of them being familial types) presented symptoms in 94.3% of the individuals as follows: bone complications (83%), renal complications (29%), and recurrent pancreatitis (11.4%) [29]. Another retrospective study on 242 individuals managed for primary hyperparathyroidism in a 24-year study of a single center confirmed that 6.19% of them (N = 15) had pancreatitis diagnosed based on two out of three of the following criteria: abdominal pain, high serum amylase (at least three times upper normal limit), or suggestive imaging features. Among this subgroup, 14/15 subjects had acute pancreatitis (93.3%), and half of them had at least one episode of recurrence; 1/15 patients had chronic pancreatitis. These individuals (N = 15) with hypercalcemia-related pancreas involvement were not associated with any other known risk factors; also, they had statistically significant higher blood calcium and alkaline phosphatase than the subgroup with primary hyperparathyroidism without any pancreas findings. A total of 14/15 of them had pancreatitis as the initial manifestation of the parathyroid disease. After parathyroidectomy, except for one subject, 14/15 patients did not experience any recurrence during a median 16-month follow-up [32]. Of note, the recurrence of pancreatitis seems rather related to post-operatory recurrent primary hyperparathyroidism rather than the pancreatic lesion per se.

Summarizing, in addition to the five previously mentioned cases of pregnancy primary hyperparathyroidism [37,38,39,41,42] and three reports of a parathyroid carcinoma [47,48,49], another seventeen articles introduced one patient per case study, except for two papers with two subjects (a total of nineteen individuals). Four children (aged between 9 and 14 years; an average age of 12; female-to-male ratio of three to one) and fifteen adults (aged between 31 and 81 years; mean age of 51.6 years; female-to-male ratio of two to one) were reported within the last five years in the field of hypercalcemia-derivate pancreatitis in patients with parathyroid NETs (N = 27 individuals) [18,21,22,25,34,35,53,54,55,56,57,58,59,60,61,62,63] (Table 4).

From a cross-disciplinary perspective, the synchronous identification of pancreatitis and primary hyperparathyroidism requires the dual management of pancreatic involvement and hypercalcemia in the sense of rapid calcium lowering, and, when clinically stable, a definite cure of primary hyperparathyroidism should be provided by parathyroidectomy. Notably, the panel of surgery indications for parathyroid tumors according to current guidelines does not distinctly stress the chapter of pancreatic complications. We may conclude that hypercalcemia-induced pancreatitis remains a treatable type during the recognition of a parathyroid NET; thus, it is mandatory to promptly and adequately manage it [64,65].

### 1.2. Multiple Endocrine Neoplasia Syndrome

An important interplay between pancreas and parathyroid status is represented by the co-presence of NETs amid MEN1 and MEN4.

#### 1.2.1. Multiple Endocrine Neoplasia Type 1 (MEN1)

##### A. Background

MEN1 (Wermer’s syndrome), an autosomal dominantly inherited syndrome with an increased penetrance, displays a poor genotype/phenotype correlation, a variation that has been hypothesized to be the effect of the “double hit hypothesis” and of a large spectrum of exogenous and endogenous factors (including epigenetic elements such as microRNAs [66] that were identified in both pancreatic and parathyroid cells) [67,68,69,70,71]. Heterozygote-inactivating pathogenic variants of the *MEN1* tumor suppressor gene (chromosome11q13) encoding the MENIN protein are followed by the somatic loss of the heterozygosity of the *MEN1* gene at the level of neuroendocrine cells [72,73]. The condition has a familial pattern in nine out of ten cases, while one out of ten subjects with MEN1 shows de novo mutations; thus, the early recognition and prompt introduction of surveillance protocols in this particular instance might not be feasible in daily practice [74,75].

The first clinical description of the syndrome came from Paul Wermer (in 1953), while the *MEN1* gene was initially sequenced in 1997 [76]. More than 1500 distinct germline and somatic pathogenic variants across the syndrome have been identified so far and continue to be identified [77,78]. In the meantime, the traditional clinical picture standing for the “three Ps”, namely pituitary, pancreas, and parathyroid NETs, extended toward a heterogonous presentation that also includes more than twenty endocrine and nonendocrine types of tumors like adrenocortical unilateral or bilateral disease (mostly adenomas, and only exceptionally carcinomas [79,80], pheochromocytoma, papillary thyroid carcinoma [81], and skin tumors such as collagenomas, lipomas/hibernomas, and angiofibromas [82], as well as cerebral tumors like meninigiomas [83], different forms of leiomyomas, and lung and thymus tumors [84,85,86,87,88]. *MEN1* pathogenic variants in exon 2, 9, and 10 are prone to a more aggressive pancreatic NET behavior [72]. Other nonendocrine components have been placed in relationship with MEN1 like breast cancer or melanoma, but currently, there is insufficient statistical evidence to directly connect them with the MEN1-associated picture [89,90].

The estimated MEN1 prevalence is two to three cases per 100,000 people with primary hyperparathyroidism being reported in 85–90% of subjects. Some MEN1 genotypes are associated with an earlier age at primary hyperparathyroidism diagnosis and a shorter time to recurrence [91]. Also, duodeno-pancreatic NETs affect between 30% and 80% of MEN1 families. Unless non-functioning (which is reported in 60–70% in some studies), almost half of them are gastrinomas (of note, hypergastrinemia-related Zollinger–Ellison syndrome should be taken into consideration [92]), and approximately one-third is represented by insulinomas; glucagonomas [93] and cystic pancreatic adenomas have been described, as well [94,95]. Overall, the clinical MEN1 onset is typically between the age of 20 and 25 years, while 95% of affected patients present neuroendocrine manifestations before the age of 50 [96].

Recently, a new concept has been released, namely, familial isolated pancreatic NETs (FIPNET), that includes subjects carrying *MEN1* pathogenic variants and a single clinical manifestation at the level of the pancreas with normal calcium/PTH levels (primary hyperparathyroidism-free phenotype). Under these circumstances, a fine index of suspicion should be considered, as well as developing a good gastroenterology/endocrinology collaborative [97]. Also, atypical scenarios are reported raising additional challenges, for instance, individuals with MEN1 manifestations but negative genetic testing, non-functioning NETs (regarding the endocrine panel) that may be more difficult to be identified early unless the carrier status is already confirmed in one individual or a family, and a very aggressive neoplasia behavior with rapid multi-organ spreading [96,98,99]. Overall, the gap between the genetic background and the expected clinical picture as well as the sporadic presentation adds more challenges to the complex panel of multiple long-term comorbidities, associated with a reduced quality of life and a general increased syndrome burden including a high mortality that comes from uncontrolled hormonal disease and metastatic tumors (mostly of pancreas origin) [76,100,101].

On the contrary, currently, particular progress in this field includes three main directions: a massive advance in imaging diagnosis including methods such as ^68^ Ga-DOTATATE PET-CT (positron emission tomography/computed tomography) to highlight the somatostatin receptors’ configuration and disease status; multilayered NET therapy, including PRRT (peptide receptor radionuclide therapy), mTOR inhibitors, somatostatin analogs, etc.; and an increased level of awareness and access for *MEN1* carriers and family members to active protocols of lifelong follow-ups that are applicable in many dedicated centers [102,103,104]. On a particular scale, the imaging exploration of parathyroid glands such as 99m-Technetium scintigraphy following the biological diagnosis of primary hyperparathyroidism might provide captures not only of the thyroid (and even myocardium) but also of the pancreas; thus, the incidental detection of a pancreatic neoplasia has been reported in localization studies before parathyroidectomy. In this particular instance, further *MEN1* gene testing is required [105,106].

Duodeno-pancreatic and thymus MEN1-NETs represent the second most common neoplasia following the presence of parathyroid NETs, meaning that many individuals confirmed with a pancreatic MEN1-NET have already been identified with primary hyperparathyroidism [107,108].

##### B. Clinical Studies

The studies’ overview (*n* = 7) showed cohorts that primarily included patients with MEN1-related insulinomas (*n* = 2) or MEN1-associated primary hyperparathyroidism (*n* = 2) or analyses of genetic testing in MEN1 (*n* = 3), a wide spectrum of objectives being registered across these data. They were all retrospective studies enrolling between 17 and 517 individuals per study, for a total of 877 MEN1 subjects [109,110,111,112,113,114,115] (Table 5).

MEN1-derivate insulinomas (N = 72 subjects) were analyzed in relationship with their outcome and the co-presence of primary hyperparathyroidism across two studies [110,111]. Zhao et al. [111] conducted a study on 55 patients with confirmed MEN1-related insulinomas (78% of them were also diagnosed with primary hyperparathyroidism); the authors showed that 69% were multifocal masses; post-surgery tumor recurrence was identified in 45% of cases; 55% of these subjects had multiple procedures of enucleation; 10% of the individuals with distal resection were associated with post-operatory pancreatic insufficiency (for a mean follow-up of 7 years) [111]. Another study on 17 subjects with MEN1-related insulinomas showed that the median age at diagnosis was 31.5 years in genetically confirmed cases (N = 7) and 69 years in the subgroup without genetic confirmation (N = 10). The rate of primary hyperparathyroidism was 6/7 and, respectively, 0/10 cases in these mentioned subgroups. Pancreatic NETs were multifocal in 3/7 or single-lesion in 10/10 cases. Metastases at diagnosis were confirmed in 3/7 and 1/10 subjects, respectively. The subgroups had a similar insulinoma size and outcome [110].

MEN1-associated parathyroid NETs (N = 585), as the most common and usually the first clinical element in MEN1, were meticulously addressed from the surgical perspective and post-parathyroidectomy outcome since this step of management is essential for an overall better prognosis. For instance, Yavropoulou et al. [112] showed in 68 patients with MEN1-related primary hyperparathyroidism that the average age at syndrome recognition was 39 ± 13.06 years and at parathyroid condition confirmation, 35.2 ± 4 years. The surgical outcome for those 82% of subjects who underwent parathyroidectomy included the following: long-term remission (56%), recurrent hyperparathyroidism (31.5%), persistent hyperparathyroidism (12.2%), and permanent hypoparathyroidism (19.2%) for a 4-year median follow-up. A total of 82% of them were associated with pancreatic NETs (71% of these tumors being hormonally non-functioning), and 66% of the individuals were identified with a pituitary NET (almost half being non-functioning). Among the persons who had a *MEN1* genetic analysis, 77% harbored pathogenic variants without genotype/phenotype correlation [112].

Moreover, a study from 2024 assessed the recurrence after initial parathyroidectomy in 517 MEN1-positive subjects confirmed with primary hyperparathyroidism (between 1990 and 2019). The first group included 187/517 individuals (34.4%) who had a prior procedure of less-than-subtotal parathyroidectomy; the second one enrolled 339/517 patients (65.5%) who were offered a subtotal parathyroidectomy. When comparing these two subgroups, the rate of recurrence was higher within the first cohort (68% versus 45%, *p* < 0.001), also associated with a shorter time to recurrence (a median of 4.25 versus 7.2 years, *p* < 0.001). *MEN1* pathogenic variants in exon 10 represented an independent risk factor concerning post-operatory recurrence; hence, once the mutation at this level is identified, less-than-subtotal parathyroidectomy should not be recommended. Santucci et al. [109] explained that despite having a high recurrence rate, re-do surgery was actually performed only in 26% of cases, and one of the main reasons, among others, was the presence of another endocrine surgery for pancreatic NETs [109].

Another study provided the genetic analysis in 40 suspected individuals for MEN1; 32/40 subjects had a syndrome confirmation, and they displayed the following spectrum: 100% had primary hyperparathyroidism; 68.7% were associated with gastro-entero-pancreatic NETs; and 66% were confirmed with a hypophyseal NET [114]. Shariq et al. [113] retrospectively analyzed 80 children with MEN1 showing that 70% (56/80) had the first manifestation before the age of 18 years at a median of 14 (ranges between 6 and 18 years, 18 being the cut-off age for the study). The profile of the parathyroid NETs included the following: the confirmation of primary hyperparathyroidism in 80% of them and 70% of them underwent a parathyroidectomy. Pancreatic and duodenal NETs were identified in 35% of the entire cohort, 70% being non-functioning NETs, 35% were insulinomas, and 5% were gastrinomas; overall, 15% of these NETs had distant metastases, and 55% of the children underwent pancreatic surgery. Pituitary NETs affected 30% of the pediatric cohort, and one-third of these were macroprolactinomas. MEN1 seemed to equally affect males and females. Genetic testing showcased 35 germline *MEN1* pathogenic variants (and 3/35 of them were novel mutations) [113].

##### C. Case Studies

At a lower level of statistical evidence, we mention the sample-focused analysis of case reports with regard to the parathyroid and pancreatic MEN1-NETs having a specified genetic confirmation of the *MEN1* pathogenic variant (regardless of if the type was detailed by the original authors) [68,73,75,77,81,97,100,116,117,118,119,120,121,122,123,124,125,126,127,128,129,130,131,132] (Table 6). 

There were 24 MEN1 subjects (including index cases with genetic analysis for their relatives) within 24 papers; they were aged between 17 years (only two pediatric cases were found) and 74 years; the female-to-male ratio was 0.84. The female population (N = 11) had an average age of 43.63 years (ranges between 28 and 60); males (N = 13) had a mean age of 40.07 years (ranges between 17 and 74) [68,73,75,77,81,97,100,116,117,118,119,120,121,122,123,124,125,126,127,128,129,130,131,132].

Five novel *MEN1* pathogenic variants have been identified as follows: missense variant in exon 3 (c.632T>A; p.Val211Asp) [97]; heterozygosity for a pathogenic insertion c.1224_1225insGTCC (p.Cys409Valfs*41) [77]; germline heterozygous *MEN1* frameshift (c.674delG; p.Gly225Aspfs*56) in exon 4 [120]; heterozygous germline pathogenic variant in exon 9 (c.1321_1323dup) [124]; and heterozygous germline in exon 5, specifically, frameshift c.930delG (NM_130799.2:c.930delG) [75]. In addition, a prior germline variant was reclassified from “uncertain significance” to “likely pathogenic” (c.1694T>A, p.L565Q) in a 74-year-old male with multiple pancreatic NETs and primary hyperparathyroidism [116].

The co-occurrence of another genetic anomaly included the following: a novel heterozygous *MEN1* germline pathogenic variant in exon 9 (c.1321_1323dup) was associated with a *CDC73* (cell cycle division) gene variant in an index case (while nine of his family members harbored similar *MEN1* pathogenic variants, and seven carried the *CDC73* variant) [124]; a 40-year-old male (and his children and two sisters) carried a *MEN1* previously known pathogen variant (c.378G>A, p.Trp126*) and glucokinase regulatory protein (GCKR) gene pathogenic variant (c.151C>T, p.Arg51*) as similarly seen only in his children [129]; a 33-year-old man was associated with a heterozygous *MEN1* gene (c.784-9G>A) and CAH-X syndrome as reflected by a homozygous complete deletion of *CYP21A2* (c.1-?_1488+? del) and a large deletion of the neighboring *TNXB* gene (c.11381-?_11524+?) in exons 35 to 44 that involved a phenotype with salt-wasting congenital adrenal hyperplasia (complicated with adrenal crisis during adulthood) and features of classic-like Ehlers–Danlos syndrome [131].

As a reinforcement of the mentioned “two hit hypothesis”, we mention a 43-year-old male with *MEN1* mosaicism (according to next-generation sequencing) harboring a mosaic pathogenic variant in exon 3 (NM_130799): c.496=/C>T, p.(Gln166=/*) in parathyroid and pancreas NETs but not in thymus NETs associated with a somatic second-hit mutation in parathyroid NETs (loss of heterozygosity) and thymus NETs, respectively (*MEN1* pathogenic variant in intron 4, c.784-9G>A) [73].

Other MEN1 phenotype specifications in this focused-analysis included with regard to primary hyperparathyroidism a pancreatic complication (acute and chronic pancreatitis) in a 56-year-old male [100]) but also a normocalcemic variant [97]. Two pancreatic NETs pointed out mostly unusual endocrine traits in terms of a growth hormone-releasing hormone (GHRH)-releasing pancreatic NET (with recurrent and metastatic behavior) in a 22-year-old male [117] and, also, in a pediatric subject diagnosed with an insulin-secreting pancreatic NET at the age of 10 which was followed by the confirmation of a GHRH-secreting pancreatic NET at the age of 18 (this represented the first pediatric case with gigantism underlying this specific pathogenic type of GH overproduction) [123].

The exceptional hormonal spectrum in the field of ACTH (Adrenocorticotropic Hormone) involved an ectopic (paraneoplastic) ACTH syndrome in a case with pancreatic and thymic NETs [121], the co-diagnosis of ACTH-independent macronodular hyperplasia (with cortisol excess) in a young lady with MEN1 [119], and bilateral adrenal tumors with subclinical Cushing’s syndrome in another adult [122].

Notably, with respect to the overall MEN1 presentation, we mention adults with two lipomas [77] or a single giant cervical lipoma [125]; for the synchronous thyroid malignancy of papillary [81] and medullary type in a *RET*-negative patient [128], two MEN1 subjects displayed a thymus carcinoid as the first MEN1 manifestation [129,132].

#### 1.2.2. Multiple Endocrine Neoplasia Type 4 (MEN4)

MEN4 (also described as “MEN1 mimicker”) [133], underlying *CDKN1B* (cyclin-dependent kinase inhibitor) germline pathogenic variants represent the rarest MEN type (an incidence of less than one individual per million-person) [134,135]. A systematic review from 2023 (which included a search between 2006, since the syndrome was first identified, and 2022) showed a total of 28 pathogenic variants of the *CDKN1B* gene associated with various clinical presentations, the majority being of missense (44%) or frameshift type (35%). Females seemed more prone to be affected (75% of all subjects) but apparently displayed a later median age at onset versus males, yet lacking statistical significance (49.5 versus 32.5 years, *p* = 0.25). Parathyroid and pituitary tumors were the most common endocrine manifestations (being confirmed in 75–76% of all the patients), whereas pancreatic NETs were diagnosed in 15% of all subjects. Of note, as opposite to MEN1, most cases of primary hyperparathyroidism were caused by a single-gland condition, not a multi-gland disease [136].

Generally, *MEN1*-negative patients should be checked for *CDKN1B* pathogenic variants (if MEN1 is suspected). Individuals with suspected familial (also called genetic or hereditary) primary hyperparathyroidism, meaning those displaying a non-MEN phenotype with a single manifestation at the parathyroid level, might harbor pathogenic variants of the *CDC73* gene [137], *CASR* (the calcium-sensing receptor), and *RET* (rearranged during transfection) genes (in MEN2) [138]. Globally, 10% of all cases confirmed with primary hyperparathyroidism display a genetic background [139,140,141].

Notably, in the study conducted by Shyamasunder et al. [114], seven out of the ten individuals who, despite being suspected with MEN1, were found *MEN1*-negative carried a p.V109G polymorphism in the *CDKN1B* gene, which required additional testing to be highlighted as pathogenic variant [114]. As prior specified, the fourth ever case of *RET*-negative medullary thyroid carcinoma in a MEN1-positive patient was reported in 2020, suggesting than a larger dichotomy in understanding this field should be kept in mind [128]. An additional case was reported in 2022 [142].

To summarize, pancreatic NETs, generally seen as a frequent type of neuroendocrine neoplasia among other sites, are associated with a genetic background in one out of ten cases, mostly MEN1 but also MEN4 and other syndromes such as neurofibromatosis type 1, Cowden syndrome, von Hippel–Lindau disease, etc. [108,143,144,145].

## 2. Pancreas-Derivate Insulin Profile with Insulin Resistance and Metabolic Syndrome Accompanying Primary Hyperparathyroidism

Patients confirmed with primary hyperparathyroidism have a higher risk of being associated with insulin resistance and metabolic syndrome and even cardiovascular mortality than the general population. The presence of insulin resistance stands for an additional turning point between the parathyroid field and pancreas involvement. Whether this specific instance is incidental or calcium- and PTH-related anomalies act as contributors to pancreas-related insulin secretion and peripheral resistance at its receptors is still an open issue [146,147].

There is evidence that high levels of calcium and low serum phosphorus (the biological aspects in primary hyperparathyroidism) alter the modulation of the insulin receptor, whereas excessive PTH concentration does not directly influence the insulin signal transduction pathway [146]. However, studies on patients without the diagnosis of primary hyperparathyroidism (such as NHANES cohort) showed that high serum PTH is positively correlated with the severity of metabolic syndrome if PTH is elevated at baseline [148].

Despite pathogenic pathways not being completely understood, many studies (not all) showed an increase in insulin sensitivity after parathyroid tumor removal (and some revealed an improvement in high blood pressure independently of insulin resistance [149]. For instance, a meta-analysis published in 2021 that included 34 studies from inception to 2020 aiming to show the impact of parathyroid tumor removal on metabolic features found no changes in the lipid profile in terms of serum triglycerides, total cholesterol, and LDL (low-density lipoprotein), cholesterol, and HDL (high-density lipoprotein) cholesterol, whereas blood levels of insulin statistically significantly decreased, as well as fasting glucose and systolic and diastolic blood pressure as opposed to the body mass index increase [150]. Notably, not all patients displayed these post-operatory cardio-metabolic dynamics, while at baseline, only a distinct subgroup of subjects had a higher prevalence of metabolic issues. Currently, the presence of metabolic syndrome in primary hyperparathyroidism does not represent an indication itself for parathyroidectomy according to guidelines [146].

When it comes to metabolic syndrome, as the signature of insulin resistance, in subjects confirmed with primary hyperparathyroidism, we identified ten studies [151,152,153,154,155,156,157,158,159,160] within the timeframe of research in addition to one study that was prior mentioned with respect to insulinoma analysis [111] and a cohort of primary hyperparathyroidism [28]. The overall results are still heterogeneous, and no clear conclusion can be drawn yet (*n* = 10, N = 1010 subjects with primary hyperparathyroidism, among them, 193 individuals had normocalcemic/mild type [152,156,158,159]). Among them, there were six longitudinal studies upon parathyroidectomy (assessments were conducted after one, three, eight, twelve, and thirteen months) [151,153,156,157,158,160], for a total of 651 patients (Table 7).

### 2.1. Cross-Sectional Analyses

In a cross-sectional study, 174 subjects with primary hyperparathyroidism (140 women and 37 males) without a previous history of chronic kidney disease or diabetes mellitus were evaluated versus 171 controls; the analysis showed that one-third of them had prediabetes according to fasting glucose levels, and 45% had insulin resistance defined as a HOMA-IR (homeostatic model assessment of insulin resistance) index > 2.6. HOMA-IR was statistically significantly higher than controls (*p* < 0.001), similarly to the results of Nomine-Criqui et al. [151]. Calcemic levels were positively correlated with HOMA-IR (r = 0.171, *p* = 0.023) and body mass index, as well (r = 0.45, *p* < 0.001), and multivariate analysis confirmed this latest mentioned parameter as being an independent predictor of insulin resistance before parathyroidectomy [155].

Another case/control study included 37 patients with primary hyperparathyroidism and a control group (N = 40) with a mean age of 51.2 years and 49.3 years, respectively. None of them had a prior diagnosis of high blood pressure, type 2 diabetes mellitus, or anomalies of the lipid profile to avoid their bias at the endothelial level. After the adjustment of other atherosclerotic factors, the intima/media thickness was higher in the studied group versus controls (*p* = 0.023), while a similar lumen diameter at the level of the carotid artery and a similar presence of endothelial plaque were identified via ultrasound [152].

### 2.2. Normocalcemic Hyperparathyroidism

A higher cardiovascular risk seems to affect individuals with the normocalcemic variant of primary hyperparathyroidism versus the healthy population despite this sub-type being generally considered to be associated with a milder clinical picture [161,162]. For example, a study of Karras et al.’s [159] compared the glucose/insulin profile of 20 individuals diagnosed with normocalcemic hyperparathyroidism and prediabetes versus controls with prediabetes and found similar levels of A1c glycated hemoglobin, HOMA-IR, serum insulin, and glucose levels at 2 h after the 75 g oral glucose test but higher fasting glucose (*p* = 0.01) that was positively correlated with PTH (rho coefficient of 0.374, *p* = 0.005) [159]. Also, a study of a small sample size (N = 14 subjects with asymptomatic primary hyperparathyroidism and normal glucose profile at baseline, mean age of 52.93 years) showed that glucagon-like peptide-1 (GLP-1) had a significant increase during the 75 g oral glucose test (*p* = 0.02 after 1 h, and *p* = 0.03 after 2 h versus baseline), probably a reflection of an early recovery phase of the glucose profile following parathyroidectomy [160]. The positive metabolic impact of parathyroidectomy in the normocalcemic type might bring an additional argument for performing surgery in these subjects which otherwise is not necessarily recommended. For example, a study followed a cohort of 16 patients for 8 months with normocalcemic hyperparathyroidism and prediabetes who underwent parathyroidectomy versus a group managed with a conservative approach (N = 16). The baseline glucose profile was similar; fasting blood glucose positively correlated with serum PTH in each group. After follow-up, the first subgroup showed an improvement when compared to the initial profile in terms of a lower fasting blood glucose (119.4 versus 111.2 mg/dL, *p* = 0.021) and glycemia level during the 2 h 75 g oral glucose test (163.2 versus 144.4 mg/dL, *p* = 0.041), while the second subgroup did not show a statistically significant glucose profile improvement [158].

### 2.3. PTH/Calcium and Insulin Secretion in Insulinomas

The insulin pattern in patients with insulinomas might be affected by the presence of high calcium and PTH, as an additional standpoint of the strong interplay of PTH/calcium/insulin. Recent data showed that hypercalcemia in patients with insulinoma in MEN1 influences the insulin pattern of secretion. As mentioned, the retrospective study on 55 Chinese patients with MEN1-related insulinoma (52% males and 47% females) showed that 78% of them had primary hyperparathyroidism, 69% were associated with pituitary tumors, and 16% were confirmed with an adrenal tumor. Insulinoma was revealed as being the first clinical element in 23.6% of the entire cohort, and one-third of them had symptomatic hypoglycemia onset before the age of 22. Subjects with hypercalcemia had lower insulin levels during hypoglycemia episodes compared to the persons with normal serum calcium (*p* < 0.001) [111].

### 2.4. Vitamin D Insights

One of the potential mechanisms involved in developing metabolic syndrome is related to low levels of vitamin D (which also might associate secondary hyperparathyroidism with the primary component) despite the fact that so far, interventional studies with vitamin D replacement have not shown the correction of metabolic traits per se [163,164]. We mention a cross-sectional study (N = 128 subjects with primary hyperparathyroidism) showing that the vitamin D-deficient subgroup (representing 51% of the entire cohort, with a serum 25-hydroxyvitamin level less than 50 nmol/L) had a statistically significantly increased prevalence of metabolic syndrome, obesity, and arterial hypertension versus the subgroup with normal 25-hydroxyvitamin D (meaning a serum level of at least 50 nmol/L). No correlation with bone and renal complications or with the originating tumor size could be established with respect to vitamin D assays [154].

### 2.5. The Impact of Parathyroidectomy on Metabolic Components

Six studies addressed the issue of metabolic changes upon parathyroidectomy, but the assessments and study design were not homogenous [151,153,156,157,158,160]. A single-centric study was performed on 65 subjects (an average age of 45.44 years) who underwent parathyroidectomy (they were assessed before surgery and one month and three months after the procedure). At the first post-parathyroidectomy evaluation, fasting blood glucose, insulin, and A1c glycated hemoglobin statistically significantly decreased when compared to baseline levels (*p* < 0.5 for each), while 92% of the subjects had a lower HOMA-IR, showing the positive impact of parathyroid removal on the insulin profile [157].

Furthermore, one single-centric prospective study conducted by Bibik et al. [153] analyzed 24 young patients aged between 18 and 50 years (median of 37; male-to-female ratio of 1 to 4) diagnosed with a parathyroid condition versus 20 healthy controls. Insulin resistance was confirmed in 54% of them. At baseline, they had increased triglyceride levels and serum insulin during both phases of secretion (hyperinsulinemic euglycemic and hyperglycemic clamps) versus controls (*p* < 0.05 for each). Statistically significant results were identified 13 months after parathyroidectomy (N = 17) with concern to lowering fasting blood glucose (*p* = 0.031) and blood insulin during its second phase (*p* = 0.039) as well as uric acid (*p* = 0.044), but no changes in the lipid profile or body mass composition according to bioelectrical impedance assessment were observed [153].

An observational longitudinal study that enrolled 231 patients with primary hyperparathyroidism (between 2016 and 2021) showed that one-third had prediabetes, and half presented insulin resistance. The individuals were followed up with for one year after parathyroidectomy. HOMA-IR statistically significantly decreased in both subgroups. The baseline predictors of post-operatory insulin resistance improvement were a HOMA-IR value above 2.5 and prediabetes according to a multivariate model [151]. Kumari et al. [28] showed that type 2 diabetic subjects (meaning 11.6%) with primary hyperparathyroidism had a lower prevalence of kidney stones when compared to nondiabetic patients confirmed with the same parathyroid condition (*p* = 0.3) and lower serum PTH (*p* = 0.04). Despite displaying an additional complication of insulin resistance, these patients showed some improved parameters; one of the explanations could be, as provided by authors, that the presence of diabetes might precipitate the early detection of high PTH due to required medical check-ups [28]

Moreover, another observational, single-center, longitudinal study between 2016 and 2019 highlighted the parathyroidectomy effect on insulin resistance. Frey et al. [156] followed subjects with traditional hypercalcemic hyperparathyroidism (N = 19) versus those with a mild form (N = 120) one year after parathyroid tumor removal. Fasting glucose and plasma insulin statistically significantly decreased within the second subgroup versus baseline (*p* < 0.001). HOMA-IR decreased in the entire cohort (*p* < 0.001). Serum triglycerides achieved a statistically significant lowering effect only within the first subgroup (*p* = 0.021) [156]. Despite being heterogeneous, the positive influence on cardio-metabolic traits should be recognized in relationship with the surgical correction of hyperparathyroidism [28,151,153,156,157,158].

## 3. Discussion

Our sample-focused analysis followed a triad of turning points between the pancreas and parathyroid with the confirmation of primary hyperparathyroidism: hypercalcemia-linked pancreatitis, *MEN1* gene pathogenic variants, and insulin resistance. These insights may be analyzed in various ways due to their complex cross-disciplinary profile. We intended to update these three pillars from the perspective of pathogenic traits, genetic testing, clinical expression in terms of NETs, and cardio-metabolic features. But the connection between the pancreas and parathyroid go way beyond the frame of primary hyperparathyroidism; for instance, we briefly discuss some additional pancreas/parathyroid turning points in order to provide an even larger picture of this topic.

### 3.1. Hypercalcemia of Malignancy in Pancreatic NETs

An inter-connection between the pancreas and hypercalcemia might involve the presence of paraneoplastic (humoral) syndrome in pancreatic NETs, either functioning (such as gastrinomas or insulinomas) or not [165]. Generally, hypercalcemia of malignancy represents the most common paraneoplastic syndrome, and it accompanies one-third of all cancers, but typically, NETs are less frequently associated with this particular endocrine complication [166].

In addition to pancreatic NET-related ACTH ectopic secretion (causing Cushing’s syndrome) [121,167], tumor-related high calcium levels are caused by PTHrP (parathyroid hormone-related peptide) overproduction that suppresses PTH levels during negative feedback via serum calcium (in patients with intact parathyroid glands) [168,169]. PTHrP encoded by the *PTHLH* gene represents the molecular signature of various tissues, including the pancreas [170]. Under physiological circumstances, it regulates pancreatic insulin secretion via its expression in beta cells. Recent murine experiments showed insulin and glucagon changes in PTHrP knockout mouse models [171].

Serum PTHrP correction upon pancreatic tumor removal might prove a useful prognostic marker in some of these cases [172]. Exceptionally, persistent hypercalcemia in patients who underwent surgery for primary hyperparathyroidism might not be a recurrent or persistent hyperparathyroidism but hypercalcemia of malignancy [173].

Another consideration is represented by developing pancreatitis in PTH-independent hypercalcemia as prior mentioned in primary hyperparathyroidism-derivate hypercalcemia. A review between inception and March 2020 showed 37 cases of paraneoplastic hypercalcemia (associated with different sites of the primary malignancy) complicated with acute pancreatitis. The average level at diagnosis was 44.8 mg/dL; one-fifth of the subjects developed necrotizing pancreatitis; one-third of the individuals died during this pancreatitis episode. Remarkably, the level of serum calcium was not correlated with disease-related mortality. On a practical basis, the prompt correction of hypercalcemia in terms of intravenous hydration and calcium-lowering agents such as intravenous bisphosphonates, calcitonin, and denosumab is mandatory to be added to standard pancreatitis management [174].

Essentially, various neoplasms, including pancreatic NETs, display liver metastasis at the moment of paraneoplastic endocrine manifestations requiring a multidisciplinary approach in terms of surgery, chemotherapy, somatostatin analogs, etc. [175]. Pancreatic adenocarcinomas might complicate too with hypercalcemia of malignancy and ectopic ACTH syndrome in association with other hormonal manifestations such as insulin-dependent diabetes mellitus as mentioned in pancreatic NETs [176]. Of note, in 2022, the first case of tumor-induced osteomalacia, another challenging cancer-related humoral syndrome underlying FGF-23 (fibroblast growth factor) excess with renal phosphate wasting effects (which was typically described in mesenchymal tumors, not necessarily malign) [177,178], was described in a 77-year-old lady confirmed with a pancreatic NET [179].

### 3.2. Autoimmune Background Affecting Endocrine Pancreas and Parathyroid

Notably, another association between endocrine pancreas anomalies and parathyroid disorders, but in the sense of hypoparathyroidism (low serum PTH and calcium), involves a common autoimmune background of the poly-glandular autoimmune syndrome, also involving type 1 diabetes mellitus (due to endocrine pancreas anomalies), chronic autoimmune Hashimoto’s thyroiditis, premature ovarian failure (hypergonadotropic hypogonadism), and Addison’s disease in different combinations affecting both children and adults [180,181,182]. Autoimmune pancreatitis represents a rare disorder caused by autoimmune reactions. Currently, two types have been reported, one with a sclerosing pattern (having a large lympho-plasmacytic infiltrate, being caused by an aberrant immune response that recently took into consideration the major role of immunoglobulin G4) and another with a duct-centric pattern that is considered idiopathic (which is not associated with a serum marker, and it requires a histological profiling for confirmation). On the other hand, autoimmune hypoparathyroidism has been reported in type 1 poly-glandular autoimmune syndrome in patients harboring AIRE pathogenic variants, usually with an early onset. This comes as an anomaly of the autoimmune regulator gene rather than direct autoimmune tissue destruction [180,181,182].

As mentioned, the case of the adult lady presenting both MEN1 and a strong autoimmune background that was published by Chavez et al. [124] in 2021 raised the issue of a potential pathogenic connection between these two syndromes; here, a novel *MEN1* pathogenic variant was identified in exon 2, while an apparently non-pathogenic variant of *CDC73* was co-present. Until further clarification, this should be regarded as incidental [124].

### 3.3. Unusual Genetic Conditions

Recently, a new case of Chanarin–Dorfman syndrome (an extremely rare condition affecting lipid profile complicated with multiple manifestations such as hepatomegaly/fatty liver, neuropathy, and deafness) was reported to be associated with endocrine issues with respect to a parathyroid lipoadenoma associated with pancreas involvement like atrophic pancreatic lesions. Less than 160 cases have been reported so far, and this 66-year-old subject was the second oldest individual diagnosed with this syndrome; he harbored a homozygous c.47+1 G>A pathogenic variant of the ABHD5 (NM_016006.6) gene [183].

### 3.4. Emergent Markers of Investigating Pancreatic and Parathyroid Conditions

Nowadays, novel molecular and immunohistochemistry markers are under development in order to highlight pathogenic insights into pancreatic and parathyroid NETs. For instance, Islet-1, a factor that interferes in epigenome programing, was found to be stain-positive in 78% of pancreatic NETs and 100% in phechromocytomas/ganglioneuromas, while its expression was negative in parathyroid tumors [184].

Alternatively, in 2023, a mouse model of monoclonal antibodies against the protein entitled calcitonin receptor-like receptor showed its physiological expression in exocrine and endocrine pancreatic cells but also in the neuroendocrine cells of pancreatic and parathyroid NETs [185].

Future research might show this protein as the target of therapy with concern to derivate tumors [185,186]. Also, at the level of murine experiments, new tracers in the field of imaging modalities are explored such as PET using radiolabeled ^18^F cinacalcet, a calcium-lowering agent that acts at the level of the calcium-sensing receptor from parathyroid glands [187]. In this particular instance, there are data showing that this G protein-coupled receptor with a major role in calcium homeostasis may activate G proteins in pancreatic cells too, not only in the parathyroid [188].

### 3.5. Endocrine Surgery

As shown, a patient might become a candidate for parathyroid and (endocrine) pancreas surgery across their life span in the particular instance of an underlying *MEN1* pathogenic variant [189]. Parathyroid surgery represents one of the most important aspects in the management of patients with primary hyperparathyroidism particularly if they already display pancreas involvement either as pancreatitis or an associated pancreas NET. The decision of parathyroid surgery should take into consideration such elements that impair the overall outcome and long-term prognosis. We mentioned above that we revised the publications during the recent COVID-19 pandemic and early post-pandemic years and highlighted the most important aspects published from a dual perspective (pancreas and parathyroid). Endocrine surgery (as similarly seen in other domains) was affected during the first waves, and a rigorous selection of patients’ admission was conducted with a potential influence of further delaying the presentation, and a more severe clinical picture was registered [190,191,192,193,194]. At this point, it is difficult to assess whether our sample-based analysis (particularly, the surgical approach and emergency presentations) might have been influenced by the data published during the COVID-19 pandemic [195].

Another aspect is related to the fact that some surgical centers are qualified for providing different types of endocrine surgery, and both sites (pancreas and parathyroid) may be mentioned across different studies performed at high-volume departments with regard to the pre-operatory presentation and post-surgery outcome (and this brings a potential bias of the present research). Hence, we mention a pediatric retrospective study from 2024 on 332 children (enrolled between 1989 and 2019) showing the data for pancreas surgery (N = 5; mean age of 14.6 ± 2.19 years; male-to-female ratio of 2 to 3; 40% of them had MEN1. Distal pancreatectomy was conducted in 4/5 cases with a post-surgery average hospitalization stay of 12 12 ± 3 days). For parathyroidectomy (N = 35; mean age of 15.2 ± 2.9 years; male-to-female ratio of 1 to 1.9; MEN1 affected 8.5% of this subgroup; median parathyroid tumor weight of 2 g) one-third of them developed hungry bone syndrome following parathyroid removal with a cure rate of 97% which is close to the general results from most studies [196].

### 3.6. Iatrogenic Interferences with Dual Hit

Notably, the general term of “endocrine malignancies” that is used in various trials might include parathyroid and pancreas neoplasms too. New drugs are under development to act against this large group of endocrine cancers. The endocrine side effects of immunotherapy currently represent a rapidly growing topic that urgently needs an in-depth study of the underlying mechanisms, long-term expected negative changes, and adequate management for the short and long term. For instance, we mention a phase II clinical trial on olaparib in patients with cancers underlying germline or sporadic pathogenic variants with homolog recombination that revealed encouraging results for *BRCA1* (breast cancer gene) and *BRCA2* genes and not for other genes such as *ATM* (Ataxia-telangiectasia mutated) or *CHEK2* (checkpoint kinase) with partial results for pancreas neoplasia and parathyroid carcinomas [197].

Furthermore, immune checkpoints inhibitors, monoclonal antibodies that increase the immune anti-tumor response, have been shown to affect endocrine glands, including the parathyroid and endocrine pancreas in addition to inducing hyper- and hypothyroidism, hypophysitis, and primary adrenal insufficiency. Type 1 diabetes mellitus due to pancreatic islets anomalies was reported in less than 1% of the subjects after a few months to a few years of drug exposure, mostly due to anti-PD-1 (programmed cell death protein) and anti-PD-L1 (programmed cell death protein ligand) antibodies rather than anti-CTLA-4 (cytotoxic T-lymphocyte associated protein). The majority of these patients had symptomatic hyperglycemia at onset, while 85% of them might complicate with ketosis. Concerning parathyroid involvement, hypoparathyroidism (and exceptionally primary hyperparathyroidism [198]) was diagnosed despite the fact that parathyroid tissue does not hold receptors for PD-1, PD-L1, or CTLA-4, but, probably, antibodies against the calcium-sensing receptor are drug-induced [199,200,201]

When exploring the fascinating cross-domain of the pancreas and parathyroid, new frontiers are revealed, and we intended to capture the essence of these insights. We are aware of the limitations related to this present work. Firstly, this is a non-systematic review, a perspective that we choose in order to provide a more flexible approach of various multidisciplinary data associated with different levels of statistical significance. Secondarily, we followed the clinical and pathogenic, including molecular and genetic, cross-disciplinary interplay between the pancreas and parathyroid rather than therapeutic strategies in each medical entity. Thirdly, the cited papers were published between January 2020 and February 2024 in order to provide the most recent perspective within the mentioned topics.

## 4. Conclusions

Our sample-focused analysis during the latest 5-year PubMed search of clinical studies and case reports followed a triad of turning points between pancreas status and primary hyperparathyroidism (hypercalcemia-linked pancreatitis, *MEN1* gene pathogenic variants, and insulin resistance) and showed a complex panel of connection elements from pathogenic factors, including biochemical, molecular, genetic, and metabolic factors, to a clinical multidisciplinary personalized approach.

## Figures and Tables

**Figure 1 ijms-25-06349-f001:**
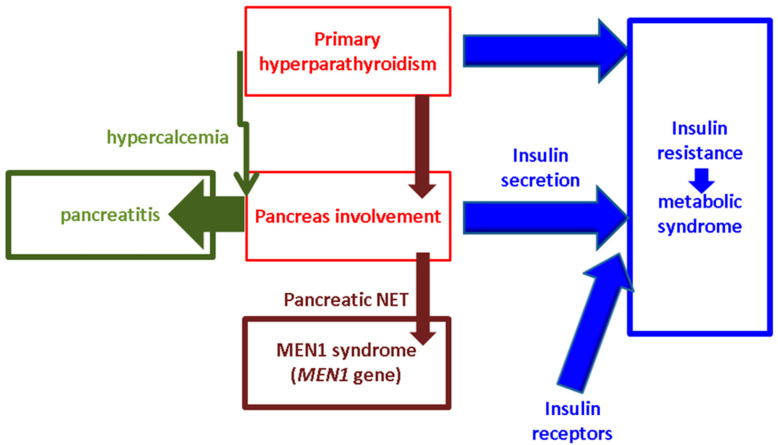
The topics of stratified research with respect to pancreas involvement (red) in subjects confirmed with primary hyperparathyroidism (red) according to our methods. The three pillars of the analysis are as follows: hypercalcemia-induced pancreatitis (green), MEN1 (multiple endocrine neoplasia) syndrome (brown), and insulin resistance (blue).

**Table 1 ijms-25-06349-t001:** Studies on patients with primary hyperparathyroidism who provided data on pancreatitis or on subjects with pancreatitis that were identified to be associated with a parathyroid NET (the table starts with the most recent publication date) [20,26,27,28,29,30,31,32].

First Author Reference Publication Year	Study Design Studied Population	Pancreas-Related Findings	Primary Hyperparathyroidism
Yattoo 2023 [26]	longitudinal (prospective) study 67 patients with chronic calcific pancreatitis	NA (for this specific case)	rate of 1/67 patients diagnosed with PHP
Rashmi 2022 [27]	retrospective study 51 patients with PHP	N = 12/51 (23.52%) with pancreatitis: N1 = (5/51) 9.8% with acute pancreatitis N2 = (7/51) 13.73% with chronic pancreatitis	N1 versus non-N1: lower age: 35.20 ± 16.11 versus 49.23 ± 14.8 years (*p* = 0.05) higher PTH: 125 versus 519.8 pg/mL (*p* = 0.01) similar serum calcium: 11.66 ± 1.15 versus 12.46 ± 1.71 mg/dL (*p* = 0.32)
Tiwari 2022 [20]	retrospective study 100 patients with acute pancreatitis	Causes of pancreatitis: 36%—biliary cause 19%—alcohol consumption 21%—idiopathic 5%—biliary + alcohol 7%—after endoscopic retrograde cholangiopancreatography 6%—drug-induced 3%—dengue infection	3% had hypercalcemia: 1% had PHP 1% had multiple myeloma 1% had hyperthyroidism
Kumari 2022 [28]	retrospective study 464 patients with PHP	N1 = 54/464 patients with 2DM Pancreatitis rate: N1 (22.2%) versus non-N1 (5.6%), *p* = 0.07	
Sharanappa 2021 [29]	retrospective study 35 pediatric subjects with PHP	11.4% had recurrent pancreatitis	8.5% had familial PHP (83% had skeletal complications; 29% had renal complications)
Arora 2021 [30]	retrospective study N1 = 30 patients with acute pancreatitis + PHP N2 = 30 controls with PHP	N1: serum calcium of 12.24 ± 2.79 mg/dL	N1: 16.6% had normocalcemia at the moment of pancreatitis
Pal 2021 [31]	registry-based study N = 386 females with PHP	50% of the females with gestational PHP had acute pancreatitis	2.1% = rate of gestational PHP
Misgar 2020 [32]	retrospective study 242 patients with PHP	N1 = 15 patients with pancreatitis (14/15—acute + 1/5—chronic calcific type)	after parathyroidectomy: 14/15 patients did not have any recurrent pancreatitis (median of follow-up: 16, ranges: 2–41 months)

Abbreviations: NA = not available; N = number of patients; PHP = primary hyperparathyroidism; PTH = parathormone.

**Table 2 ijms-25-06349-t002:** Case reports of gestational primary hyperparathyroidism complicated with acute pancreatitis during gestation or the postpartum period according to our methods of research (the table starts with the most recent publication date) [37,38,39,41,42].

First Author Reference Publication Year	Studied Population	Pancreas-Related Findings	Primary Hyperparathyroidism
Musleh 2023 [37]	31-year-old female gravida 4, para 3 30th week of gestation	acute pancreatitis	parathyroid adenoma
Niu 2023 [38]	26-year-old pregnant female	maternal eclampsia → cesarean → postpartum life-threatening hypercalcemia + acute pancreatitis	symptomatic medical therapy → planned surgery→ post-operatory confirmation of a parathyroid adenoma
Tsai 2021 [42]	31-year-old pregnant female (week 37)	acute pancreatitis during pregnancy	parathyroid adenoma first presentation during pregnancy
Dias Leite 2020 [41]	40-year-old primigravida	postpartum acute pancreatitis	two parathyroid adenomas (pre-eclampsia due to primary hyperparathyroidism)
Bansal 2020 [39]	30-year-old pregnant female	acute pancreatitis during second trimester of pregnancy	cystic parathyroid adenoma

**Table 3 ijms-25-06349-t003:** Case reports of parathyroid carcinoma complicated with pancreatitis according to our methods of research (the table starts with the most recent publication date) [47,48,49].

First Author Reference Publication Year	Studied Population	Pancreas-Related Findings	Primary Hyperparathyroidism
Zelano 2022 [48]	56-year-old male	moderate pancreatitis	parathyroid carcinoma Among other clinical elements: portal thrombosis kidney stones brown tumors
Mignini 2021 [47]	56-year-old male	pancreatic pseudocyst due to acute pancreatitis	parathyroid carcinoma Bone and peritoneal metastases that were first misdiagnosed as pancreatic cancer
Jiajue 2020 [49]	34-year-old female	recurrent acute pancreatitis amid recurrent post-operatory PHP	mediastinal parathyroid carcinoma (recurrent post-parathyroidectomy for primary hyperparathyroidism)

**Table 4 ijms-25-06349-t004:** Reported data (case studies) on pancreatitis in patients with PTH-related hypercalcemia during the diagnosis of primary hyperparathyroidism according to our research (the table starts with the most recent publication date) [18,21,22,25,34,35,53,54,55,56,57,58,59,60,61,62,63].

First Author Reference Publication Year	Study Design	Studied Population	Pancreas-Related Findings	Primary Hyperparathyroidism	Other Key Findings
Thakker 2023 [34]	case report	55-year-old female	acute pancreatitis	cystic parathyroid adenoma	first admission for diabetic ketoacidosis +papillary thyroid carcinoma +hypertrophic obstructive cardiomyopathy
Sarwal 2023 [56]	case report	56-year-old male	necrotizing pancreatitis	nephrocalcinosis acute kidney injury	
Mehta 2023 [55]	case report	52-year-old female	3 episodes of pancreatitis within 6 months	parathyroid adenoma (complicated with brown tumors)	
Arhoun El Haddad 2022 [22]	case report	41-year-old male	acute pancreatitis (pseudocystic appearances after acute episode)		
Fu 2022 [54]	case report	57-year-old female	recurrent pancreatitis	parathyroid adenoma	
Singh 2022 [57]	case report	35-year-old female	acute persistent pancreatitis with a pancreatic pseudocyst	giant left inferior parathyroid adenoma	
Ali 2022 [58]	case report	12-year-old female	acute pancreatitis	parathyroid adenoma	
Oh 2022 [59]	two case reports	9-year-old female 14-year-old male	pancreatitis in both cases	parathyroid adenoma in both cases (ureter stone in both cases)	
Desmedt 2021 [25]	case report	81-year-old female	acute pancreatitis	parathyroid adenoma asymptomatic hypercalcemia	
Yang 2021 [21]	case report	56-year-old female	acute pancreatitis due to a double cause	primary hyperparathyroidism + hypercalcemia of malignancy	zoledronate + calcitonin to control hypercalcemia → exploratory parathyroidectomy → chemotherapy + surgery for poorly differentiated uterine cancer → serum calcium decrease
Bin Yahib 2021 [18]	case report	13-year-old female	recurrent abdominal pain within 8 months (pancreatitis)	thymic parathyroid adenoma	
Sudharshan 2021 [60]	case report	65-year-old male	acute pancreatitis	parathyroid adenoma	
Kuchay 2021 [61]	two case reports	35-year-old female 56-year-old female	severe acute necrotizing pancreatitis acute pancreatitis	parathyroid adenoma parathyroid adenoma	
Mills 2021 [35]	case report	31-year-old male	acute necrotizing pancreatitis		onset with severe diabetic ketoacidosis
Loderer 2021 [62]	case report	67-year-old male	acute edematous pancreatitis	parathyroid hyperplasia	
Rahou 2020 [53]	case report	52-year-old female	acute calcific pancreatitis	parathyroid adenoma	multiple endocrine neoplasia type 1 (+pituitary and corticoadrenal adenoma)
Kundu 2020 [63]	case report	35-year-old female	recurrent episodes of acute pancreatitis (after gallstone-induced pancreatitis corrected with cholecystectomy)	parathyroid adenoma	

**Table 5 ijms-25-06349-t005:** Clinical studies introducing MEN1-related pancreatic NETs associated with specific data regarding the presence of primary hyperparathyroidism (within our timeframe and methods of research) [109,110,111,112,113,114,115].

First Author Reference Publication Year	Study Design Studied Population	Pancreatic Involvement	Parathyroid Involvement
Santucci 2024 [109]	retrospective study 517 patients with recurrent MEN1-related PHP N1 = 187/517 (34.4%) patients with less than subtotal parathyroidectomy N2 = 339/517 (65.5%) patients with subtotal parathyroidectomy	re-do parathyroid surgery was conducted in 26% due to interferences such as pancreatic NET surgery	N1 versus N2: rate of recurrence: 68% versus 45%, *p* < 0.001 time to recurrence: 4.25 versus 7.2 years, *p* < 0.001 independent predictor of recurrence: pathogenic variants in exon 10
Opalińska 2023 [110]	retrospective study 17 patients with MEN1-related insulinomas	N1 = 7 patients with *MEN1* genetic confirmation MEN1 N2 = 10 patients without genetic confirmation N1 had multifocal insulinomas (3/7) metastasis at diagnosis: 3/7 (N1) versus 1/10 (N2)	rate of PHP: 6/7 (N1) versus 0/10 (N2)
Zhao 2022 [111]	retrospective study 55 patients with MEN1-related insulinomas	69% of cases were multifocal insulinomas	78% of cases associated with PHP
Yavropoulou 2022 [112]	retrospective study 68 patients with MEN1-related PHP	82% of cases had pancreatic NETs (71% of these tumors were non-functioning)	age at PHP diagnosis: 35.2 ± 4 years 82% underwent surgery with following outcomes (4-year follow-up): 56% long-term remission 31.5% recurrent hyperparathyroidism 12.2% persistent hyperparathyroidism 19.2% permanent hypoparathyroidism
Shariq 2022 [113]	retrospective study 80 pediatric patients with MEN1	35% pancreatic and duodenal NET (55% underwent surgery) endocrine profile: 70% non-functioning NETs 35% insulinomas 5% gastrinomas 15% had metastases	80% had PHP (70% underwent parathyroidectomy)
Shyamasunder 2021 [114]	retrospective genetic study 40 suspected patients with MEN1 (32/40 had a confirmation)	68.7% associated gastro-entero-pancreatic NETs	100% had PHP
Pieterman 2021 [115]	retrospective genetic study N1 = 39 genotype-negative patients with clinical MEN1 N2 = 63 genotype-positive MEN1 index cases	multifocal duodeno-pancreatic NETs: 17% (N1) versus 68% (N2)	PHP with single-gland involvement: 47% (N1) versus 0% (N2)

Abbreviations: N = number of patients; MEN = multiple endocrine neoplasia; PHP = primary hyperparathyroidism.

**Table 6 ijms-25-06349-t006:** Case studies (one patient per article) addressing the clinical presentation in MEN1 patients diagnosed with pancreatic NETs and primary hyperparathyroidism and having a genetic confirmation of the *MEN1* pathogenic variant (according to our methods of literature research) [68,73,75,77,81,97,100,116,117,118,119,120,121,122,123,124,125,126,127,128,129,130,131,132].

Case Number First Author Reference Publication Year	Studied Patient (Index Case) Genetic Testing of *MEN1* Gene—Pathogenic Variants Affected Family Members (If Any)	Pancreatic Involvement ^#^ Parathyroid Involvement Other Key Findings
Kalshetty 2023 [100]	56-year-old male Pathogenic variant: exon 10 (segregation analysis) Son: insulinoma Father: gastric malignancy	Metastatic duodeno-pancreatic NET Prior surgery for a parathyroid adenoma (PHP) History of recurrent kidney stones Prior episode of acute pancreatitis Chronic pancreatitis (also related to alcohol consumption) + diabetes mellitus
Horikoshi 2023 [97]	40-year-old female Novel germline pathogenic variant: missense variant in exon 3 (c.632T>A; p.Val211Asp) Mother: multiple pancreatic NETs	Multiple pancreatic NETs Normocalcemic PHP
Matsubayashi 2023 [116]	74-year-old male Germline variant (reclassified from “uncertain significance” to “likely pathogenic) (c.1694T>A, p.L565Q) Son: PHP + multiple pancreatic NETs Daughter: PHP	Multiple gastroenteric and pancreatic NETs PHP
Molina 2023 [77]	37-year-old male Novel germline—heterozygosity for a pathogenic insertion c.1224_1225insGTCC (p.Cys409Valfs*41) (direct sequence analysis) 2/6 tested family members were found positive	Insulinoma PHP (prior history of kidney stones) Two lipomas
Xu 2022 [81]	35-year-old female Confirmation of *MEN1* pathogenic variant (no other specification) Father: gastric tumor	Pancreatic and gastro-duodenal NET PHP (parathyroid cysts and hyperplasia) Papillary thyroid carcinoma Pituitary non-functioning microadenoma
Koivikko 2022 [117]	22-year-old male c.466G>C, p.(Gly156Arg) pathogenic variant (similar to his mother)	Growth hormone-releasing-hormone releasing pancreatic NET (recurrent and metastatic) PHP (recurrent after surgery)
Petriczko 2022 [118]	17-year-old male Confirmation of *MEN1* germline pathogenic variant Father: PHP (including acute pancreatitis) Sister: PHP + ganglioglioma + pancreatic NET	Insulinoma PHP
Lagarde 2022 [73]	43-year-old male *MEN1* mosaicism (next-generation sequencing): Mosaic pathogenic variant in exon 3 of *MEN1* (NM_130799): c.496=/C > T, p.(Gln166=/*)—confirmed in parathyroid and pancreas but not in thymus tumors Somatic second-hit mutation in parathyroid: loss of heterozygosity Somatic second-hit mutation in the thymus tumor: *MEN1* pathogenic variant in intron 4, c.784-9G>A	Multiple pancreatic NETs PHP Thymus NET
Chavoshi 2022 [119]	32-year-old female Heterozygote for *MEN1* pathogenic variant: 645,773,330-64577333AGAC, c.249-252delGTCT, p. (11e85Serfs Ter33) in exon 2	Insulinoma PHP ACTH-independent macronodular hyperplasia (cortisol excess)
Welsch 2022 [120]	28-year-old female Novel germline heterozygous *MEN1* frameshift mutation (c.674delG; p.Gly225Aspfs*56) in exon 4 Father: PHP + non-secretor pancreatic NET + pituitary adenoma brother: PHP + non-secretor pancreatic Sister: PHP non-secretor pancreatic NET + prolactinoma	Insulinoma PHP Prolactinoma
Li 2022 [121]	49-year-old male Confirmation of *MEN1* pathogenic variant (no other specification)	Pancreatic NET PHP Thymic NET Ectopic ACTH syndrome Diabetes insipidus at onset due to hypercalcemia + hypokalemia
Koyama 2022 [122]	51-year-old female Missense germline *MEN1* pathogenic variant (c.1013>G, p.Leu338Pro) in exon 7 (DNA sequencing)	Non-functioning pancreatic NETs PHP (parathyroid cysts) Prolactinoma Adrenal tumors with subclinical Cushing’s syndrome
Srirangam Nadhamuni 2021 [123]	18-year-old male Heterozygous *MEN1* germline mutation (c.249_252delGTCT, p.I85Sfs (similar to his father)	Insulin-secreting pancreatic NET at 10 years Growth hormone-releasing hormone-secreting pancreatic NET (first pediatric case with gigantism) at 18 years PHP (surgery at 15 years)
Chavez 2021 [124]	50-year-old female Novel heterozygous germline pathogenic variant in exon 9 of the *MEN1* gene (c.1321_1323dup) +“benign” CDC73 gene variant Multiple family carriers (of *MEN1* = 19; of *CDC73* = 7)	Metastatic gastric type 1 NET Recurrent PHP Pernicious anemia Mucosa-associated lymphoid tissue lymphoma of the thyroid Positive anticalcium sensor receptor antibodies BRCA 1/2-negative invasive breast cancer
Fushimi 2021 [125]	28-year-old male Germline frameshift (c.1613delA) in exon 10	Multiple fatty deposits in the pancreas PHP Macroprolactinoma Giant cervical lipoma
Cho YY 2021 [126]	32-year-old female Frameshift germline pathogenic variant, NM_130799.1:c.1546dupC (p.Arg516Profs∗15) (similar to her daughter)	Multiple pancreatic NETs PHP Prolactinoma
Gilis-Januszewska 2021 [127]	54-year-old female *MEN1* LRG_509 c.781C>T (p.Leu261Phe) missense variant in exon 4 5/6 family members: PHP 2/6 family members: pancreatic NETs	Aggressive pancreatic NETs (insulinoma + non-functioning) PHP
Boro 2020 [68]	46-year-old female Deletion: c.824-832delGGTACCCCA in exon 5 (whole-gene sequencing) Father: insulinoma + PHP Brother: multiple insulinomas + PHP + acromegaly Other 8 members of her family with at least one MEN1 component	Insulinoma (well-differentiated pancreatic NET G1) PHP (parathyroid hyperplasia) (recurrent kidney stones for prior 2 decades) Prolactinoma
Naruse 2020 [75]	52-year-old female Novel frameshift c.930delG *MEN1* heterozygous germline (NM_130799.2:c.930delG) mutation in exon 5	Non-functional pancreatic NET PHP (single parathyroid tumor) Pituitary non-functioning microadenoma
Friziero 2020 [128]	60-year-old female Confirmation of *MEN1* sporadic pathogenic variant (no other specification) + negative *RET* mutation	Non-functioning pancreatic NET Recurrent PHP (parathyroid hyperplasia) Medullary thyroid carcinoma
Zheng 2020 [129]	40-year-old male *MEN1* pathogenic variant (c.378G>A, p.Trp126*) (+the same for his children and his two sisters) Mutations of GCKR gene (c.151C>T, p.Arg51*) (+the same in his children)	Pancreatic cancer Insulinoma with intrahepatic metastasis PHP Thymic carcinoid (first MEN1 manifestation)
Demirtaş 2020 [130]	64-year-old male Frameshift pathogenic variant in exon 10 (c.1680_1683delTGAG) (whole-gene Sanger sequencing) 17/25 relatives had MEN1 11/18 tested relatives had the same mutation (**)	Pancreatic NET/Prevalence in positive kindred (**) 16/17 (94.1%) PHP/Prevalence in positive kindred (**) 5/17 (29.4%) Pituitary tumor/Prevalence in positive kindred (**) 5/17 (29.4%)
Chen Cardenas 2020 [131]	33-year-old male Heterozygous *MEN1* gene (c.784-9G>A) +CAH-X syndrome (***) as reflected by the following: homozygous complete deletion of *CYP21A2* (c.1-?_1488+? del) (complete deletion of *CYP21A2* gene) +Large deletion of the neighboring *TNXB* gene (c.11381-?_11524+?) (large deletion of *TNXB* gene in exons 35 → 44)	Pancreatic NET PHP Prolactinoma (***) Salt-wasting congenital adrenal hyperplasia (complicated with adrenal crisis) (***) Classic-like Ehlers–Danlos syndrome primary hypogonadism
Ma 2020 [132]	40-year-old male Pathogenic variant in *MEN1* gene c.378G>A (p.Trp126*) Sister: prolactinoma	Pancreatic NET G2 PHP Prolactinoma Thymic carcinoid (first MEN1 manifestation)

Abbreviations: ACTH = Adrenocorticotropic Hormone; BRCA = breast cancer gene; CAH-X syndrome = contiguous gene deletion syndrome of congenital adrenal hyperplasia and Ehlers–Danlos syndrome; CDC73 = cell cycle division; G = grading; GCKR = gene of glucokinase regulatory protein; NET = neuroendocrine tumor; MEN = multiple endocrine neoplasia; PHP = primary hyperparathyroidism; **^#^** these are the terms used for the original diagnosis within the cited papers; ** the same family; *** elements of CAH-X syndrome.

**Table 7 ijms-25-06349-t007:** Original studies regarding pancreas-related insulin profile and associated cardio-metabolic traits, including metabolic syndrome in patients with primary hyperparathyroidism (articles published on PubMed between January 2024 and January 2020) [28,111,151,152,153,154,155,156,157,158,159,160].

First Author Reference Publication Year	Study Design	Studied Population	Outcome/Endpoint
Nomine 2024 [151]	observational, longitudinal 1-year study after parathyroidectomy	N = 231 patients with PHP N1 = 75 patients with prediabetes (fasting glucose ≥ 1 g/L) N2 = 108 patients with insulin resistance (HOMA > 2.5)	After parathyroidectomy: N1: lower HOMA versus pre-surgery (*p* = 0.04) N2: lower HOMA versus pre-surgery (*p* = 0.001)
Elbuken 2023 [152]	case/control study	N1 = 37 patients with normocalcemic PHP without HBP, DM or dyslipidemia (mean age of 51.2 ± 8 years) N2 = 40 controls (mean age of 49.3 ± 7.5 years)	intima/media thickness at carotid artery: N1 > N2 (0.65 mm versus 0.59 mm, *p* = 0.023)
BibiK 2023 [153]	longitudinal (prospective) 13-month study after parathyroidectomy	N1 = 24 patients with PHP (median age of 37) N2 = 20 controls	N1: 54% with insulin resistance At baseline: N1 > N2: triglycerides (*p* < 0.05) N1 > N2: serum insulin during both phases of secretion (hyperinsulinemic euglycemic and hyperglycemic clamps) (*p* < 0.05 for each) After parathyroidectomy: N1: lower fasting blood glucose versus pre-surgery (*p* = 0.031) N1: lower insulin during second phase of secretion (*p* = 0.039) N1: similar triglycerides versus pre-surgery (*p* < 0.5)
Şengül Ayçiçek 2023 [154]	cross-sectional study	128 subjects with PHP	51% with 25-hydroxyvitamin D < 50nmol/L (N’) N’ versus non-N’: higher prevalence of metabolic syndrome (*p* = 0.04), obesity (*p* = 0.01), and high blood pressure (*p* = 0.03)
Zhao 2022 [111]	retrospective study	55 patients with MEN1-related insulinoma 78% had PHP	patients with hypercalcemia (N = 24) versus patients with normal calcium (N = 31) had lower insulin during hypoglycemia episodes (*p* < 0.001)
Al-Jehani 2022 [155]	cross-sectional study	N1 = 174 patients with PHP without DM, and CKD N2 = 171 controls	N1: 36% with prediabetes + 45% with insulin resistance (HOMA > 2.6) HOMA: N1 > N2 (3.386 ± 3.111 versus 1.919 ± 1.158; *p* < 0.001)
Frey 2022 [156]	observational, longitudinal 1-year study after parathyroidectomy	N1 = 19 patients with classic PHP N2 = 120 patients with mild PHP	After parathyroidectomy: N1: lower triglycerides versus pre-surgery (*p* = 0.021) N2: lower fasting glucose and plasma insulin versus pre-surgery (*p* < 0.001) N1 + N2: lower HOMA (*p* < 0.001)
Kumari 2022 [28]	retrospective single-center study	464 patients with PHP N1 = 54 (11.6%) patients had 2DM	N1 versus non-N1: lower PTH (203 versus 285 pg/mL, *p* = 0.03) lower rate of kidney stones (18.5% versus 36.1%, *p* = 0.03)
Nikooei Noghani 2021 [157]	observational, longitudinal 1-month and 3-month study after parathyroidectomy	N = 65 patients with PHP (mean age of 45.44 ± 9.59 years)	After 1 month since parathyroidectomy: a reduction in fasting blood glucose (*p* < 0.5) serum insulin (*p* < 0.5) A1c glycated hemoglobin (*p* < 0.5) HOMA decrease in 92% of the patients
Karras 2020 [158]	observational, longitudinal 8-month study after parathyroidectomy (N1) versus conservative approach (N2)	N1 = 16 patients with normocalcemic PHP and prediabetes N2 = 16 patients with normocalcemic PHP and prediabetes with similar glucose profile versus N1 at baseline	After 8 months since parathyroidectomy (N1): lower fasting blood glucose (119.4 ± 2.8 versus 111.2 ± 1.9 mg/dL, *p* = 0.021) lower glycemia during 2 h 75 g oral glucose test (163.2 ± 3.2 versus 144.4 ± 3.2 mg/dL, *p* = 0.041) After 8 months of conservative management (N2): similar glycemia during 2 h 75 g oral glucose test (167.2 ± 2.7 versus 176.2 ± 3.2 mg/dL, *p* = 0.781)
Karras 2020 [159]	cross-sectional study	N1 = 20 patients with normocalcemic PHP and prediabetes N2 = 20 controls with prediabetes	N1: fasting blood glucose—PTH correlation: rho = 0.374, *p* = 0.005 N1 versus N2: higher fasting blood glucose (105.6 ± 2.8 versus 98.2 ± 1.8 mg/dL, *p* = 0.01)
Antonopoulou 2020 [160]	observational, longitudinal after parathyroidectomy	N = 14 patients with asymptomatic PHP and normal glucose profile	After parathyroidectomy: GLP-1 increase during oral glucose test after 1 h (63 ± 44.7 versus 102.6 ± 40.2 pg/mL, *p* = 0.02) and after 2 h (71 ± 35.9 versus 102.49 ± 40 pg/mL, *p* = 0.03) when compared to pre-operatory level

Abbreviations: CKD = chronic kidney disease; DM = diabetes mellitus; GLP-1 = glucagon-like peptide-1; HOMA = homeostatic model assessment of insulin resistance; HBP = high blood pressure; MEN = multiple endocrine neoplasia; N = number of patients; PHP = primary hyperparathyroidism.

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
