# Peer review of "Turning Points in Cross-Disciplinary Perspective of Primary Hyperparathyroidism and Pancreas Involvements: Hypercalcemia-Induced Pancreatitis, MEN1 Gene-Related Tumors, and Insulin Resistance"

_ijms, 2024, doi:10.3390/ijms25126349_

Round 1

Reviewer 1 Report

Comments and Suggestions for Authors

Thank you for a very comprehensive review. 

It is very long (probably too long), especially the paragraph about hypercalcemia and MEN-1. 

I would suggest that you either use more Tables or - if data are presented in Tables - that you remove these data from the text. Further, it is also difficult to comprehend why certain studies need to be mentioned in the text. 

Further, I would propose to make the aim more clear. Which aim(s) were there, and in the Discussion or Conclusion - you answer them clearly. Right now the Discussion and Conclusion need to be rewritten so your important points with this review are presented more clearly. 

Could you rephrase "Plus, we mention..." in line 708 in paragraph 3.5?

Is it a knocking mouse model or a knock out mouse model? (Discussion 4.1)

Author Response

Response to Review 1 Comments

Dear Reviewer,

Thank you very much for your time and your effort to review our manuscript.

We are very grateful for providing your valuable feedback on the article.

Here is our response and related amendment that has been made in the manuscript according to your review (marked in yellow color).

Thank you for a very comprehensive review. 

Thank you very much.

It is very long (probably too long), especially the paragraph about hypercalcemia and MEN-1. 

Thank you very much. We respectfully mention that the spectrum of data is extremely large and our perspective stands for molecular, genetic, hormonal and practical points. MDPI rules do not limit the length of one paper. We follow your recommendation and revisited the paragraph. Also, to make it clearer, we split the section into 3 parts: background (section 2.2.1.A.), clinical studies (section 2.2.1.B.) and data coming from case studies (section 2.2.1.C.). Thank you

I would suggest that you either use more Tables or - if data are presented in Tables - that you remove these data from the text. Further, it is also difficult to comprehend why certain studies need to be mentioned in the text. 

Thank you very much. We already introduced seven tables. The studies that are particularly mentioned bring clarity to prior controversies or represent  an upper level of statistical evidence than prior (to our time frame of research) data or they pose new topics and new highlights that is why we consider that the data/parameters and the main outcome are essential. Thank you

Further, I would propose to make the aim more clear. Which aim(s) were there, and in the Discussion or Conclusion - you answer them clearly. Right now the Discussion and Conclusion need to be rewritten so your important points with this review are presented more clearly. 

Thank you very much. We followed your recommendations and extended the presentation of the aim of this work. “Across this comprehensive review, we aimed to provide an in-depth analysis with respect to various pancreas involvements in patients diagnosed with parathyroid NETs, particularly those functioning types causing primary hyperparathyroidism. Specifically, there are three main aspects to present: hypercalcemia that causes pancreatitis amid these parathyroid tumors, pancreatic and parathyroid NETs co-presence and the pancreas involvement as essential contributor to insulin resistance that is identified in subjects with primary hyperparathyroidism.” At Discussion further expansion of this topic included differential diagnosis of the hypercalcemia amid other malignancies of different origins, a potential new topic to address a common autoimmune background for pancreas and parathyroid glands, new genetic configurations for both glands, new molecular markers, the importance of endocrine surgery and potential iatrogenic interference for both organs.  Thank you

Could you rephrase "Plus, we mention..." in line 708 in paragraph 3.5?

Thank you very much. We corrected it. Thank you

Is it a knocking mouse model or a knock out mouse model? (Discussion 4.1)

Thank you very much. We corrected it. Thank you

Thank you very much.

Reviewer 2 Report

Comments and Suggestions for Authors

Comments:

  1. Clarity and Depth of Autoimmune Associations Comment: Quite interesting insight is given in section 4.2 related to auto-immune interactions between pancreas and parathyroid glands regarding the shared auto-immune mechanisms that are common to both the glands and may affect either of the two glands. However, it could be elaborated that immunological pathways and how they exactly modulate gland function would add rationalization to it. Much clearer distinction would be gained between direct autoimmune destruction versus secondary effects caused by autoimmune dysregulation in other glands.
  2. Genetic Syndromes and Conditions Covered Comment: The author brings about some of the other issues on the MEN1 and Chanarin-Dorphman syndromes, indicating their effects on the pancreas and the parathyroid. As much as these references point out the clarification, they would make a more structured and in-detail analysis in the article. More exact detail on the penetrance and expressivity of such genes and their interaction with other genetic or environmental factors in contributing to the pathophysiology of associated endocrine disorder would be valuable. Furthermore, the discussion of novel MEN1 pathogenic variants may be extended to discuss their possible implications for treatment or, in the light of this genetic information, prevention.
  3. Human Innovations in Diagnostic Markers Comment: The section on novel molecular and immunohistochemical markers (section 4.4) is a good start, but it's time for some depth here. Data should also include specificity and sensitivity, compared with current diagnostic tools, for them to be considered viable for use in a clinical setting. This section would come off very well in describing the clinical relevance of such markers for diagnosis, prognosis, and monitoring of therapy.
  4. Surgical Consideration Iatrogenic Comment: This is of immense relevance in the respect of surgical nature related to most treatments of disorders of pancreas and parathyroides in the context of implications for endocrine surgery and iatrogenic effects (sections 4.5 and 4.6). In other words, a more critical risk-to-benefit analysis of surgery in various contexts (e.g., MEN1 vs. non-MEN1 conditions) is needed. In addition, the endocrine side effect of immunotherapy is a rapidly growing area that urgently needs an in-depth review of the mechanisms, potential long-term effects, and strategies of managing conditions evoked by treatment.
  5. Research Gaps and Methodological Rigor Comment: The article is quite generally describing the relationship between the pathologies of the pancreas and parathyroid. It would likely have failed to give a systematically literate review and fail to benefit from a more strictly methodological framework. Future studies with suggested research gaps and future directions that can identify and address this missing link would make this part more human.
  6. Interdisciplinary Approach Comment: The article is very much to the point in suggesting a multidisciplinary approach, but actually, it offers no insight on how the integration of such disciplines could be realized in practice. Very possibly, case studies in detail or clinical scenarios where such approaches have been applied for success would provide practical insights into interdisciplinarity, which help implement such strategies. Purpose Review Overall, the paper does present at the outset a wide-eyed perspective of detailed interplays of the pancreas and parathyroid glands. Deeper genetic analyses and much elaboration on diagnostic and therapeutic innovations, along with a stronger methodological framework, may substantially increase its impact and relevance to the field.

Author Response

Response to Review 2 Comments

Dear Reviewer,

Thank you very much for your time and your effort to review our manuscript.

We are very grateful for your insightful comments and observations, also, for providing your valuable feedback on the article.

Here is a point-by-point response and related amendments that have been made in the manuscript according to your review (marked in yellow color).

  1. Clarity and Depth of Autoimmune Associations Comment: Quite interesting insight is given in section 4.2 related to auto-immune interactions between pancreas and parathyroid glands regarding the shared auto-immune mechanisms that are common to both the glands and may affect either of the two glands. However, it could be elaborated that immunological pathways and how they exactly modulate gland function would add rationalization to it. Much clearer distinction would be gained between direct autoimmune destruction versus secondary effects caused by autoimmune dysregulation in other glands.

Thank you very much. We followed your recomandation and extended the section. “Autoimmune pancreatitis represents a rare disorder caused by autoimmune reactions. Currently, two types have been reported, one with sclerosing pattern (having a large lympho-plasmacytic infiltrate; being caused by an aberrant immune response that recently took into consideration the major role of immunoglobulin G4) and another with involve a duct-centric pattern is considered idiopathic (which does not associate a serum marker, and it requires a histological profiling for confirmation). On the other hand, autoimmune hypoparathyroidism has been reported in type 1 poly-glandular autoimmune syndrome in patients harboring AIRE pathogenic variants, usually with an early onset. This comes as an anomaly of the autoimmune regulator gene rather than direct autoimmune tissue destruction”. Thank you

  1. Genetic Syndromes and Conditions Covered Comment: The author brings about some of the other issues on the MEN1 and Chanarin-Dorphman syndromes, indicating their effects on the pancreas and the parathyroid. As much as these references point out the clarification, they would make a more structured and in-detail analysis in the article. More exact detail on the penetrance and expressivity of such genes and their interaction with other genetic or environmental factors in contributing to the pathophysiology of associated endocrine disorder would be valuable. Furthermore, the discussion of novel MEN1 pathogenic variants may be extended to discuss their possible implications for treatment or, in the light of this genetic information, prevention.

Thank you very much. We respectfully mention that the genetic implications have been mentioned with respect to the aim of this work, but further genetic – treatment interferences are out of the scope of this work which is already considered too long. Thank you

  1. Human Innovations in Diagnostic Markers Comment: The section on novel molecular and immunohistochemical markers (section 4.4) is a good start, but it's time for some depth here. Data should also include specificity and sensitivity, compared with current diagnostic tools, for them to be considered viable for use in a clinical setting. This section would come off very well in describing the clinical relevance of such markers for diagnosis, prognosis, and monitoring of therapy.

Thank you very much. Currently, these markers are not practical for everyday use, neither for the clinicians that treat patients with primary hyperparathyroidism and associated pancreas involvement. At this point they represent future topics to be explored and we do not have enough data for in-deep analysis with a practical/clinical relevance. Thank you

  1. Surgical Consideration Iatrogenic Comment: This is of immense relevance in the respect of surgical nature related to most treatments of disorders of pancreas and parathyroides in the context of implications for endocrine surgery and iatrogenic effects (sections 4.5 and 4.6). In other words, a more critical risk-to-benefit analysis of surgery in various contexts (e.g., MEN1 vs. non-MEN1 conditions) is needed. In addition, the endocrine side effect of immunotherapy is a rapidly growing area that urgently needs an in-depth review of the mechanisms, potential long-term effects, and strategies of managing conditions evoked by treatment.

Thank you very much. We followed your recommendations and extended the section.  “Parathyroid surgery represents one of the most important aspects in the management of patients with primary hyperparathyroidism particularly if they already display a pancreas involvement either as pancreatitis or an associated pancreas NET. The decision of parathyroid surgery should take into consideration such elements that impair the overall outcome and the long term prognosis”. Also, we added your excellent suggestions with regard to new drugs as following “The endocrine side effects of the immunotherapy currently represents a rapidly growing topic that urgently needs an in-depth study of the underlying mechanisms, long-term expected negative changes and adequate management for short and long term.” Thank you

  1. Research Gaps and Methodological Rigor Comment: The article is quite generally describing the relationship between the pathologies of the pancreas and parathyroid. It would likely have failed to give a systematically literate review and fail to benefit from a more strictly methodological framework. Future studies with suggested research gaps and future directions that can identify and address this missing link would make this part more human.

Thank you very much. This is a non-systematic review. This narrative review as study design allows a more flexible type of approach due to the heterogeneity of the spectrum in primary hyperparathyroidism with regard to the pancreas involvement; that is why, we choose to introduce the data as a narrative review since various levels of statistical evidence and various themes/topics and data are identified in the mentioned papers. On the other hand, a systematic review pinpoints a specific critical assessment which in the matter of pancreas and primary hyperparathyroidism is less feasible (with regard to a wide, not a limited area of interplays and crossroads). However, this type of review is a well-recognized, standard, and traditional approach which is suitable for topics such as the update of the most recent data on primary hyperparathyroidism-associated pancreas issues. This allowed us to examine and evaluate the scientific panel on this specific topic in various aspects.  In addition, the limitations of the current work are introduced at the end of Discussion section. Thank you

  1. Interdisciplinary Approach Comment: The article is very much to the point in suggesting a multidisciplinary approach, but actually, it offers no insight on how the integration of such disciplines could be realized in practice. Very possibly, case studies in detail or clinical scenarios where such approaches have been applied for success would provide practical insights into interdisciplinarity, which help implement such strategies. Purpose Review Overall, the paper does present at the outset a wide-eyed perspective of detailed interplays of the pancreas and parathyroid glands. Deeper genetic analyses and much elaboration on diagnostic and therapeutic innovations, along with a stronger methodological framework, may substantially increase its impact and relevance to the field.

Thank you very much. We approach the topic of pancreas and parathyroid at the level of molecular markers, endocrine issues, pathological reports and immunohistochemistry analysis, tumor aspects, practical points, genetic testing, surgical approach and outcome. To our aware this is a cross-disciplinary perspective which in daily clinical and surgical practice stands for different practitioners and specialists including at the level of excellence centers. Thank you

Thank you very much.
